# E-Cigarette Liquid Provokes Significant Embryotoxicity and Inhibits Angiogenesis

**DOI:** 10.3390/toxics8020038

**Published:** 2020-05-27

**Authors:** Anas A. Ashour, Hashim Alhussain, Umar Bin Rashid, Labiba Abughazzah, Ishita Gupta, Ahmed Malki, Semir Vranic, Ala-Eddin Al Moustafa

**Affiliations:** 1College of Medicine, QU Health, Qatar University, Doha 2713, Qatar; aa1510042@qu.edu.qa (A.A.A.); ur1608109@qu.edu.qa (U.B.R.); la1605015@qu.edu.qa (L.A.); svranic@qu.edu.qa (S.V.); 2Biomedical Research Center, Qatar University, Doha 2713, Qatar; h.alhussain@qu.edu.qa (H.A.); ishugupta28@gmail.com (I.G.); 3Biomedical Science Department, College of Health Sciences, QU Health, Qatar University, Doha 2713, Qatar; ahmed.malki@qu.edu.qa

**Keywords:** e-cigarettes, embryo, angiogenesis, toxicity, gene deregulations

## Abstract

E-cigarette smoking (ECS) is a new method of tobacco smoking that is gaining popularity as it is thought to be a “healthy method” of tobacco consumption. The adverse outcomes of ECS on the respiratory and cardiovascular systems in humans have been recently demonstrated. Nevertheless, the effect of e-cigarette liquid (ECL) on the early stage of embryogenesis and angiogenesis has not been explored yet. Chicken embryo at 3 days of incubation and its chorioallantoic membrane (CAM) of 5 days were used to explore the outcome of ECL on the embryo. Real-time PCR was also employed to study the regulation of a set of key controller genes of embryogenesis as well as angiogenesis. Our study revealed that ECL exposure is associated with a high rate of mortality in embryos as around 70% of treated embryos, at 3 days of incubation, die after 5 days of exposure. Additionally, ECL inhibits angiogenesis of the CAM of 5 days of incubation by more than 30%. These effects could be explained by the upregulation of *ATF-3, FOXA2, INHBA, MAPRE-2*, and *RIPK-1,* as well as the downregulation of *SERPINA-4* and *VEGF-C* genes, which are important key controller genes of embryogenesis as well as angiogenesis. Our data suggest clearly that ECS can have dramatic toxic outcomes on the early stage of embryogenesis as well as angiogenesis. Accordingly, we believe that further studies to assess the effects of ECS on human health are essential.

## 1. Introduction

According to the World Health Organization [1], tobacco smoking accounts for 8 million deaths worldwide. It is estimated that there are 1.1 billion users of tobacco products worldwide accounting for 32% of the male and 7% of the female population [2]. There are various methods of tobacco consumption with the most common ones being cigarettes, electronic cigarettes (e-cigarettes), cigars, and water-pipe smoking. It has been well established that smoking causes a plethora of diseases including chronic obstructive pulmonary disease, cardiovascular events, rheumatoid arthritis, osteoarthritis, as well as multiple types of cancer such as upper respiratory, lung, bladder, renal, and pancreatic cancer [3].

E-cigarettes are new electronic devices that heat a nicotine or non-nicotine containing liquid with flavors stored in a refillable cartridge into a vapor to be inhaled [4]. E-cigarette smoking (ECS) was intensely advertised when released and was marketed as a healthier alternative to traditional cigarettes and a healthier device to switch to when quitting traditional smoking. Moreover, there are no clear legislations that restrict the use of ECS, such as imposing taxes, as it is not clearly defined. Additionally, in some countries ECS is not included in tobacco smoking products. Thus, it is not surprising that their prevalence is becoming higher, particularly among young adults [5,6]. Due to the absence of tobacco combustion, many of the toxins of traditional cigarettes such as carbon monoxide are absent. However, e-cigarette vapors have been known to contain low concentrations of some tobacco-derived chemicals such as carcinogenic nitrosamines, formaldehyde, and acrolein [7]. Furthermore, the fluid and vapor of e-cigarettes have been shown to contain heavy metals and silicate particles [8]. Moreover, materials used in flavoring and byproducts of the production process are associated with some serious adverse side effects [9]. Due to these reasons, questions have been raised on the health impact of e-cigarettes. Several reports have implicated e-cigarettes in common respiratory disorders such as acute bronchiolitis, acute eosinophilic pneumonia, and bilateral pleural effusions [10]. Furthermore, the aerosol produced by e-cigarettes is known to contain carcinogenic compounds such as nitrosamines and heavy metals. Consequently, concerns have also been raised on the cancer risk of long-term ECS. However, there is no definitive evidence suggesting a risk being present. It has also been shown that e-cigarettes may pose a cardiovascular risk, particularly to those who already have cardiovascular problems [11]. Lastly, second-hand exposure to e-cigarette vapors may also pose a risk to individuals according to a systematic review that has shown that second-hand exposure to e-cigarette smoke has the potential to lead to adverse health effects [12]. Most of these health risks are thought to be associated with ECS’ effect on the inflammatory process, systemic dysfunction, and gene deregulation [13,14].

Although they have been anticipated as a safer substitute to conventional cigarettes, e-cigarettes are largely consumed by pregnant women. Thus, a few studies have investigated the impact of e-cigarettes on mice offspring. It was revealed that ECS during pregnancy adversely affects mice offspring, especially lung health and cognitive function [15,16]. With this in mind, studying the effect of e-cigarettes using a chicken embryo model could provide valuable information on the embryotoxicity of ECS as well as its effects on angiogenesis and embryos’ survival, given the fact that no previous studies have investigated these effects on the chicken embryo model and angiogenesis before. Hence, the goal of this study is to investigate the outcome of e-cigarette liquid (ECL) on the early stages of the embryo and angiogenesis by exploring its effects on survival and the normal development of blood vessels in addition to a specific set of genes related to these biological events.

## 2. Materials and Methods

### 2.1. E-Cigarette Liquid

We used ECL as it is a convenient approach to obtain preliminary information about the toxic role of ECS, as ECL was used in other studies related to the outcome of ECS on the embryo [17]. ECL known as “Virginia Tobacco” with a nicotine concentration of 6 mg/mL in 70%, 30%, and 15% of propylene glycol (organic compound), vegetable glycerin, and fruit flavor, respectively, (E-Lixir, Montreal, QC, Canada) was chosen, since “Virginia tobacco” of 6 mg/mL nicotine is one of the most popular ECLs used by young people in Canada and Qatar. The liquid was stored in a dark cabinet at a temperature of 4 °C as recommended by the manufacturer. Based on our previous work, we used a final concentration of 2 mg/mL of nicotine [17,18,19]; the liquid was mixed with PBS (Qiagen, Toronto, ON, Canada) to obtain the desired concentration. This study was approved by the Institutional Biohazard Committee of Qatar University (QU-IBC) on 22 May 2019 (number: 2019/032).

### 2.2. Embryos and In Ovo E-Cigarette Treatment

A total of 97 White Leghorn chicken embryos were fertilized and incubated in similar conditions. Embryos were either treated with 30 or 60 μg suspension of final nicotine concentration from ECL dissolved in PBS. The final concentrations of 30 and 60 μg of nicotine were used based on our recent investigation of water-pipe smoking on embryo and cancer cells as well as other investigations related to the outcome of ECL on the embryo [17,18,19]. Control embryos were treated with PBS alone. From these embryos, 25 were treated at day 5 of incubation and observed for 48 h before being examined for the effect of ECL on angiogenesis; while, 72 embryos were exposed at day 3 of incubation and observed for another 5 days for survival analysis. All embryos that survived till 8 days of age were sacrificed. Autopsies from brain and heart tissues were taken for RNA extraction to perform RT-PCR analysis.

### 2.3. Macroscopic and Microscopic Analysis

Chicken embryos treated with ECL were examined under the stereomicroscope and compared with their matched controls. Angiogenesis of the chorioallantoic membrane (CAM) and the embryo was observed, and images were captured. Moreover, embryonic tissues were also inspected under the microscope with a special focus on brain, heart, and liver tissues as previously described [18,20].

### 2.4. RNA Isolation and Reverse Transcription (RT)-PCR Analysis

Total RNA was purified from brain, heart, and liver tissues of chicken embryos using the RNeasy Plus Mini Kit (Qiagen, Valencia, CA, USA) according to the manufacturer’s instructions. cDNA synthesis and PCR amplification were performed using Invitrogen Superscript III One-Step RT-PCR System with Platinum™ Taq DNA Polymerase (Thermo Fisher Scientific, Mississauga, ON, Canada) according to the manufacturer’s protocol.

RT-PCR amplification was performed using primer sets for activating transcription factor-3 (*ATF-3*), forkhead box-A2 (*FOXA2*), inhibin beta-A (*INHBA*), microtubule-associated protein RP/EB family member-2 (*MAPRE-2*), receptor-interacting serine-threonine kinase-1 (*RIPK-1*), serpin peptidase inhibitor-4 (*SERPINA-4*), and vascular endothelial growth factor-C (*VEGF-C*) (Table 1).

*GAPDH* primers were used as an internal control. In order to obtain a relative quantification of gene expressions, images acquired from RT-PCR were analyzed using ImageJ software 1.52k (National Institute of Mental Health, Bethesda, MD, USA.) [21]. The intensity of the bands relative to the *GAPDH* bands were used to calculate a relative expression of genes in each of the analyzed tissues (brain, heart, and liver).

### 2.5. Angiogenesis Quantification, Statistical and Survival Analysis

To test the difference in angiogenesis, three parameters (vessel area, number of junctions, and total vessel length) were quantified using AngioTool program version 0.6a as described elsewhere [22].

A simple 2 × 2 chi square table was used to compare the difference in survival between subjects treated with e-cigarettes and their matched controls. Kaplan–Meier survival analysis curves for the two groups were plotted and compared using log-rank test. Normal distribution of all measurements was tested using histograms and the Shapiro–Wilk test. Means of normally distributed data were compared using T-test, while non-normally distributed data were compared using Mann–Whitney U-test. Statistical Package for Social Sciences (SPSS) 64-bit version 23 was used to carry out the aforementioned tests. All tests were two-tailed, and results were considered statistically significant if *p*-values were <0.05.

## 3. Results

A total of 72 chicken embryos were treated at 3 days of incubation with either ECL or clear buffer solution and observed for 5 days in order to explore the outcome of ECL on survival during embryonic development. The embryos were divided based on the treatment concentration of ECL into 60 μg, 30 μg, and control with 30, 31, and 11 embryos, respectively. A significant difference was observed in the survival as 76% of embryos treated with 60 μg ECL died, compared to 64% in the 30 μg ECL group and 9% in the control group (*p* < 0.001; Table 2).

Kaplan–Meier survival analysis was used to compare the number of dead embryos. Most of the embryos in the treatment groups died in the first 24 h of treatment especially in the higher concentration group. This significant difference was also evident when comparing the survival curves of the groups using log-rank test (*p* = 0.001; Figure 1).

Microscopic and macroscopic examination of surviving embryos was performed with a special focus on heart, brain, and liver tissues. No morphological changes were observed in the treatment group. Additionally, samples were taken from these tissues to examine the expression of key regulator genes of embryogenesis. RT-PCR was performed to explore the effects of ECL on the gene expression patterns of *ATF-3, FOXA2, INHBA, MAPRE-2, RIPK-1, SERPINA-4*, and *VEGF-C* in brain, heart, and liver tissues of the embryos. Our results showed that *ATF-3, FOXA2, INHBA, MAPRE-2*, and *RIPK-1* were upregulated in ECL-treated embryos compared to their matched controls. On the other hand, *SERPINA-4* and *VEGF-C* genes were downregulated in the ECL-treated subjects. These results were statistically significant and consistent in all three different tissues (Figure 2).

Angiogenesis of the CAM and the embryo was examined in 25 chicken embryos (8 exposed to 60 μg ECL and 12 to 30 μg ECL in addition to 5 control samples). Embryos were exposed at the 5th day of incubation as we were aiming for optimal vascularization time. The embryos were examined 24 and 48 h after exposure. The embryos showed a decrease in all assessed parameters in a dose–effect manner. The number of junctions and total vessel length were significantly less in embryos treated with low or high concentration of ECL. Conversely, the vessel area was significantly less with high ECL concentration but not with the lower concentration (*p* = 0.052, Figure 3).

## 4. Discussion

Despite the rapidly increasing worldwide consumption of e-cigarettes, their safety remains largely unproven. Hence, the current study aimed to investigate the effects of ECL on early embryogenesis using a chicken embryo model. To our knowledge, no studies investigated the effects of ECL on embryo survival and its effects on the early stages of embryogenesis including angiogenesis. Therefore, we decided to use the chicken embryo model as it has been used to study similar parameters in previous studies [18,23]. Our study revealed that exposure to chemicals in e-cigarettes significantly decreased the survival likelihood in chicken embryos as 76% of chicken embryos in the 60 μg ECL group died compared to only 11% from their matched controls. This finding is consistent with the current literature on the effects of traditional cigarette smoking on embryo survival [23]. Other studies have elucidated the harmful effects of ECL on embryogenesis albeit by using different animal models. It was shown that exposure to e-cigarette aerosols inhibited cardiac transcription factors and hence cardiac development in a zebrafish model and in human embryonic stem cells [24]. It was also shown that exposing zebrafish embryos to 1,2 propanediol, which is a major component of e-cigarettes, increased the incidence of string heart and yolk sac edema [25].

Currently, there is no consensus on the exact toxic mechanisms involved in ECS-related systematic dysfunction and cellular damage. Nonetheless, recent experimental and clinical data shows that ECS increases inflammation, inflammatory mediators, and oxidative stress [13,14]. The data also suggest that ECS may potentiate mechanisms for initiating cellular dysfunction as well as significantly altering genetic expression and enhancing mutations [26]. Human lung cells, for instance, were shown to develop oxidative toxicity and morphological changes in response to aerosols produced by vaporizing e-cigarette liquids in several studies [13,14]. Other investigations have also reported cytotoxic effects on cultured cardiomyoblasts. Interestingly, it was found that this effect is mainly associated with the byproducts used in flavorings rather than with nicotine itself [9]. In addition, exposure to e-cigarette smoke is reported to result in an impaired immunological response [27]. Additionally, a recent investigation conducted on respiratory epithelium has shown that e-cigarettes might lead to a decreased expression of immune-related genes similar to that caused by cigarettes and water-pipe smoking through inactivation of transcription factors, such as Early Growth Response 1 (EGR1) [27]. Other studies have shown that ECS affects the immune system by impairing pulmonary antibacterial and antiviral defenses. In in vivo experiments, mice that were exposed to e-cigarette liquid vapor elicited severely defective pulmonary bacterial clearance after being infected with *Streptococcus pneumoniae*. Reduced phagocytosis by alveolar macrophages is thought to be the key event in the pathogenesis of airway diseases [28]. Moreover, our study showed that ECL negatively affected the angiogenesis in the CAM of ECL-treated embryos in a dose–effect manner. Our findings may be explained by the increase in oxidative stress and inflammation known to be caused by ECS in exposed embryos [29]. These findings implicate the extensive toxic effects of ECL exposure on embryogenesis and suggest that expecting mothers should be warned of the effects ECS could have. This is also consistent with relevant literature studying the effects of traditional cigarettes on angiogenesis in embryos [30].

Lastly, our study demonstrated that ECL exposure also increased gene expression of *ATF-3, FOXA2, INHBA, MAPRE-2*, and *RIPK-1* and decreased *SERPINA-4* and *VEGF-C* in the hearts, brains and livers of chicken embryos. Thus, it is clear that ECL can affect the physiology of exposed organisms via the deregulation of key genes and modulation of essential transcription factors such as *ATF-3,* which represents one of the key regulators of cell proliferation [31], as well as *RIPK-1,* which regulates cellular death and apoptosis [32]. Furthermore, downregulation of *VEGF-C* expression could explain the effect of ECL on angiogenesis. These data are consistent with our previous work on water-pipe smoking and single-walled carbon nanotubes (SWCNTs) on embryogenesis where we demonstrated the dramatic effects and cytotoxicity of carbon and nicotine on similar genes and how they affect embryogenesis [18,20]. In fact, only few data are available on the effects of ECL on gene expression. A study conducted by Canistro et al. (2017) described the toxicological and mutagenic effects of e-cigarettes on genes, which may lead to cancer. It was discovered that e-cigarettes damage DNA not only at the chromosomal levels as characterized by strand breaks, but also at a genetic level since they were shown to cause point mutations, particularly among younger consumers [26]. Furthermore, they also found that e-cigarettes possess a potent promoting effect on phase-I carcinogen-bioactivating enzymes, including activators of polycyclic aromatic hydrocarbons, in addition to DNA oxidation to 8-hydroxy-2’-deoxyguanosine [26].

The ECL generally consists of nicotine, flavorings, and a humectant [33]. Although the harmful health effects of nicotine are well known, it is believed that its effects are nascent when delivered as an aerosol [33]. Furthermore, it has been indicated that inhaling ECL and holding it for a longer time, helps to develop a stronger vape; however, this increases the amount of pulmonary deposition resulting in higher drug absorption in the blood as well as toxicity [34]. A previous study, evaluated the presence of toxins in both ECL and aerosols; they found the presence of diacetyl and acetyl propionyl in large quantities in ECL, thus, exposing users to higher than recommended safety levels of these toxins [35]. Moreover, in comparison to cigarette smoke extract, toxic effect of ECL is comparatively less [9,36,37]. In addition, ECL was used to test its effect on inflammatory response and innate immune defense against human rhinovirus infection in vitro and in vivo [38]. Additionally, another in vivo study in asthmatic mice models used ECL to study its effect on respiratory health; after 10 weeks, ECL elevated pro-inflammatory cytokine levels and airway hyper-responsiveness to methacholine challenge [39]. The health effects observed in the study were due to increased sensitivity of “asthmatic” lungs to inhaled respiratory irritants and not solely due to ECL effect. These observations led us to use ECL over aerosol and hence the embryos were treated with ECL [39].

Although e-cigarettes contain less amount of nicotine, we believe that other components available in ECL, especially flavorings, contribute to the cytotoxic effect of ECL demonstrated by our study. This is supported by the study published by Omaiye et al. (2019) that assessed the effect of eight commonly used flavorings in different concentrations and showed a concentration–response toxic effect of all types [40]. Furthermore, heavy metals such as Mn, Ni, and Zn in ECL could potentially contribute to this toxicity [8]. Finally, volatile organic compounds such as acrylamide could play a role in toxicity since it was shown to be a reproductive toxicant [41].

## 5. Conclusions

In our study, we reveal, for the first time, that ECL has dramatic adverse effects on the early stage of the normal embryonic development. This effect, which includes hindered angiogenesis and even death, is mainly due to the effect of ECL on the key regulator genes of embryogenesis that are *ATF-3, FOXA2, INHBA, MAPRE-2, RIPK-1, SERPINA-4,* and *VEGF-C*. Although our study clearly defines the toxic outcomes of ECL, more studies are needed to further elucidate the mechanism by which ECS affects embryogenesis and other aspects of human pathologies, as well as the embryonic response to ECL using non-lethal doses.

## Figures and Tables

**Figure 1 toxics-08-00038-f001:**
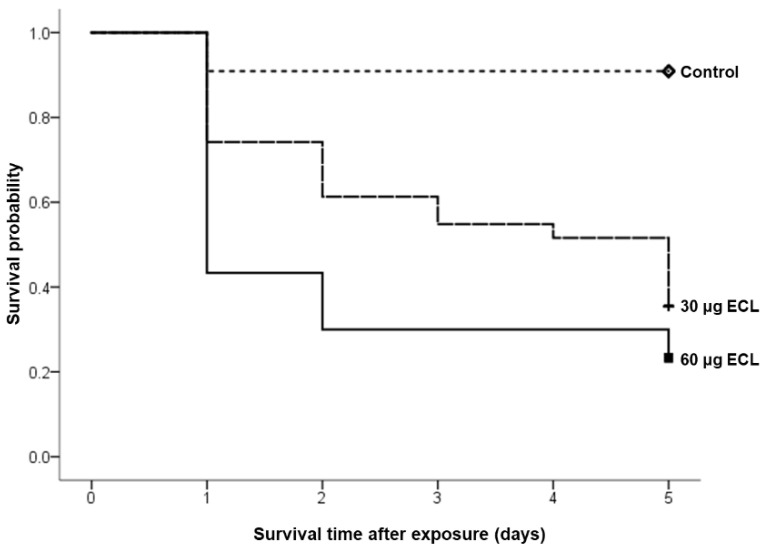
Survival analysis of ECL-treated embryos and their matched controls. It is clear that ECL significantly decreases the survival probability of exposed embryos in comparison with their control (*p* = 0.001).

**Figure 2 toxics-08-00038-f002:**
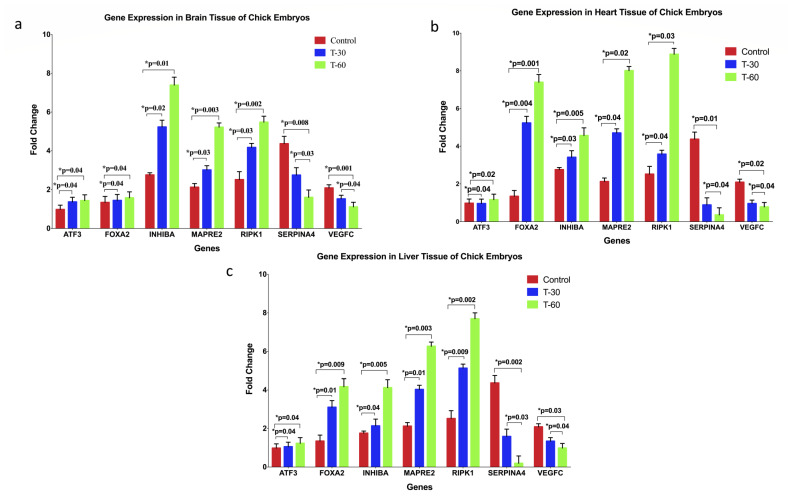
RT-PCR analysis of *ATF-3, FOXA2, INHBA, MAPRE-2, RIPK-1, SERPINA-4*, and *VEGF-C* genes in brain (**A**), heart (**B**), and liver (**C**) of chicken embryos at 8 days of incubation. We observed that ECL induced an upregulation of *ATF-3, FOXA2, INHBA, MAPRE-2*, and *RIPK-1* and downregulation of *SERPINA-4* and *VEGF-C* in comparison with control tissues. The embryos were exposed to ECL at 3 days of incubation as described in the Materials and Methods section.

**Figure 3 toxics-08-00038-f003:**
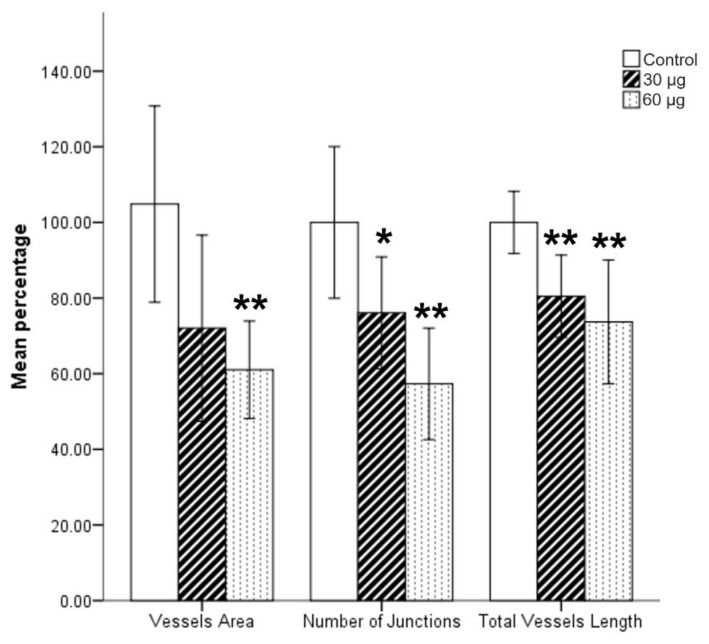
Quantification of chorioallantoic membrane (CAM) angiogenesis of chicken embryos at seven days of incubation using AngioTool program. Blood vessel area, number of junctions, and total vessel length were compared between ECL-treated CAMs and their matched controls. It is clear, from this analysis, that ECL inhibits angiogenesis of the CAM in ECL-treated embryos in comparison with untreated ones. Embryos were exposed to ECL at 5 days of incubation as illustrated in the Methods section. Total samples, *n* = 25; * *p* < 0.05; ** *p* < 0.01.

**Table 1 toxics-08-00038-t001:** Primer sets for *ATF-3, FOXA2, INHBA, MAPRE-2, RIPK-1, SERPINA-4*, and *VEGF-C* used for RT-PCR amplification.

Gene Name	Primers
***ATF-3***	5′ - AAAAGCGAAGAAGGGAAAGG - 3′
	5′ - ATACAGGTGGGCCTGTGAAG - 3′
***FOXA2***	5′ - GACCTCTTCCCCTTCTACCG - 3′
	5′ - AGGTAGCAGCCGTTCTCAAA - 3′
***INHBA***	5′ - GCCACCAAGAAACTCCATGT - 3′
	5′ - GCAACGTTTTCTTGGGTGTT - 3′
***MAPRE-2***	5′ - CAAAGGAGCCTTCCACAGAG - 3′
	5′ - GTCACTTCTGATGGCAGCAA - 3′
***RIPK-1***	5′ - CCGTACAGAATTGCAGCAGA - 3′
	5′ - TTCCATTAGCACACGAGCTG - 3′
***SERPINA-4***	5′ - CCAGCAAAAGGGAAAATGAA - 3′
	5′ - CACCACTGATGCCAGAGAGA - 3′
***VEGF-C***	5′- AGGGAACACTCCAGCTCTGA - 3′
	5′ - CTCCAAACTCTTTCCCCACA - 3′
***GAPDH***	5′ - CCTCTCTGGCAAAGTCCAAG - 3′
	5′ - CATCTGCCCATTTGATGTTG - 3′

**Table 2 toxics-08-00038-t002:** Number of embryos used in survival analysis and percentage that died after 5 days. We note that approximately 70% of e-cigarette liquid (ECL)-treated embryos die 5 days after treatment. The embryos were exposed to ECL at 3 days of incubation as described in the methods section. Chi-square test was used for comparison (*p* < 0.001).

Embryos	Number of Cases	Number of Embryos Dead Before 8 Days of Incubation (%)
**60** **μ** **g ECL**	30	23 (76)
**30** **μ** **g ECL**	31	20 (64)
**Control**	11	1 (9)

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
