# Peer review of "E-Cigarette Liquid Provokes Significant Embryotoxicity and Inhibits Angiogenesis"

_toxics, 2020, doi:10.3390/toxics8020038_

Round 1

Reviewer 1 Report

The goal of the present manuscript is to investigate the effect of e-cigarette liquid (ECL) on early embryogenesis and angiogenesis using check embryo. The manuscript addresses a significant question regarding the safety of e-cigarette during pregnancy. However, several issues need to be addressed:

  1. It was not clear why the authors decided to use ECL liquid not EC smoke or EC extract which is more clinically relevant as embryos do not get directly exposed to ECL. Additional justification and clinical significance of using ECL should be addressed.
  2. The rational of picking ECL “ Virginia tobacco” is not addressed. Is it one of the commonly used e-liquid by pregnant female?
  3. On what basis the authors picked 2mg/ml as final nicotine concentration? What is the clinical relevance of treatment with 30 ug or 60 ug suspension? Does it resemble real life scenarios of e-cigarette exposure?
  4. The discussion would benefit from adding a paragraph about the potential toxicant of ECL that may contribute of the observed phenotype.

Minor:

  1. In figure 2 b and c, is the difference between control and T-30 for ATF3 expression statistically different? It looks like there is no difference.
  2. Figure 3 is missing the asterisk of statically significance on the figure itself.
  3. Add the number of mice used to the legends of the figures (n= ).

Author Response

Reviewer comment:  It was not clear why the authors decided to use ECL liquid not EC smoke or EC extract which is more clinically relevant as embryos do not get directly exposed to ECL. Additional justification and clinical significance of using ECL should be addressed.

Our Response: We thank the reviewer for his/her critical comments that have helped us to improve the paper.

We used ECL as a convenient way to get some preliminary information about the toxic role of EC smoking. In addition, ECL was used in other studies related to the outcome of EC on the embryo (Raez-Villanueva et al., 2018). This has been mentioned in section 2.1 (page 3, lines 91-93). On the other hand, it is well established today that smoking can affect the normal development of the human embryos. Thus, and as we mentioned, this methodology and animal model is a very suitable approach as a first step to exhibit the toxic outcome of such an addiction. The use of chicken embryo in the study is a very well-known model to study embryogenesis including embryotoxicity, which is why we opted to use it.

Reviewer comment:  The rational of picking ECL “Virginia tobacco” is not addressed. Is it one of the commonly used e-liquid by pregnant female?

Our Response: Virginia tobacco of 6 mg/ml nicotine is the most popular ECL in Canada and Qatar used by young people, there are no specific information about the percentage of pregnant female consumption of ECL, nevertheless, in this type of studies, second-hand smoking can be as dangerous as direct consumption. Thus, we used this type of ECL in our investigation. This has been addressed in the methodology (please refer to the E-cigarettes liquid section).

Reviewer comment:  On what basis the authors picked 2mg/ml as final nicotine concentration? What is the clinical relevance of treatment with 30 ug or 60 ug suspension? Does it resemble real life scenarios of e-cigarette exposure?

Our Response: Based on our previous work and other work on embryo and cancer cells, a concentration of 2mg/ml was chosen as the final nicotine concentration (Ashour et al., 2018; Sadek et al., 2018; Raez-Villanueva et al 2018). This is now included in section 2.1 of the Materials and Methods section. The final concentrations of 30ug and 60 ug of nicotine was also used based on our recent investigation of water-pipe smoking on the embryo and cancer cells as well as other investigations related to the outcome of ECL on the embryo (Ashour et al., 2018; Sadek et al., 2018; Raez-Villanueva et al 2018) and has been included in section 2.2 of the Materials and Methods section. An EC device is usually filled with one ml of ECL, which is consumed in one or two sessions, this liquid is then heated, thus breaking up the liquid to several toxic components including carbon monoxide and others. Also, we have added a sentence in the conclusion about the need of further studies to be conducted using some other non-lethal concentrations.

Reviewer comment:  The discussion would benefit from adding a paragraph about the potential toxicant of ECL that may contribute of the observed phenotype.  

Our Response: The paragraph has been added at the end of the Discussion section as recommended.

Reviewer comment:  In figure 2 b and c, is the difference between control and T-30 for ATF3 expression statistically different? It looks like there is no difference.

Our Response: Due to the wide range of the Y-axis of the graph (up to 10 folds). The relatively small, yet significant difference in the ATF3 expression is obscured. However, the numbers showed it as statistically significant (p=0.04) (page 5).

Reviewer comment:  Figure 3 is missing the asterisk of statically significance on the figure itself. Add the number of mice used to the legends of the figures (n=).

Our Response: The asterisk has been added to the figure to denote the significance and the number of embryos used in the study has been included in the figure legend as suggested by the reviewer (page 6).

Reviewer 2 Report

The authors of the manuscript endeavor to investigate the impacts of electronic cigarettes use on embryonic development by means of the exposure of chick embryos exposed to Liquids used in E-cigarettes. Minor revision recommended.

  1. Citation number 8 should be replaced with a better quality citation. Do not use citation 7 for both purposes. A number of labs with no experience in inorganic analysis have entered the hot "yapping" field and published poor quality data. The Williams, ... Talbot group is one of these. The use of acid in glass vessels for collection of aerosol for metals analysis renders their data completely unreliable.
  2. Citation 7 is acceptable for the comment supported regarding toxic organic constituents of e-cigarette aerosols, but should not be used as a substitute for citation 7, since Goniewicz et al. also use glass in sample preparation for metals analysis. The report of cadmium in 11 of 14 aerosols analyzed by this group shows that their metals data is also unreliable.
  3. The authors should comment in Discussion, that electronic cigarette liquid was used for these experiments and not the aerosol, so this could make a difference in the embryonic responses.
  4. The authors should consider with regard to comment 3 to what extent these results are relevant to human embryonic development, since human embryos are not directly exposed as are chick embryos.
  5. Since the majority of chick embryos did not survive 5 days of exposure, the authors should comment on the need for other experiments on embryonic responses at nonlethal doses.

Author Response

Reviewer comment:  Citation number 8 should be replaced with a better quality citation. Do not use citation 7 for both purposes. A number of labs with no experience in inorganic analysis have entered the hot "yapping" field and published poor quality data. The Williams, ... Talbot group is one of these. The use of acid in glass vessels for collection of aerosol for metals analysis renders their data completely unreliable.

Our Response: We thank the reviewer for his/her constructive comments that have helped us to improve the exposition of the manuscript. As suggested, the citation was replaced with a better-quality research assessing metal components (page 2, line 59).

Reviewer comment:  Citation 7 is acceptable for the comment supported regarding toxic organic constituents of e-cigarette aerosols, but should not be used as a substitute for citation 7, since Goniewicz et al. also use glass in sample preparation for metals analysis. The report of cadmium in 11 of 14 aerosols analyzed by this group shows that their metals data is also unreliable.

Our Response: The reference 8 has been replaced by Olmedo et al., 2018 (page 2, line 59).

Reviewer comment:  The authors should comment in Discussion, that electronic cigarette liquid was used for these experiments and not the aerosol, so this could make a difference in the embryonic responses. The authors should consider with regard to comment 3 to what extent these results are relevant to human embryonic development, since human embryos are not directly exposed as are chick embryos.

Our Response: In the discussion, the reason has been given for choosing electronic cigarette liquid over aerosol as stated in the last two paragraphs of the Discussion section (pages 7-8, lines 242-263).

Reviewer comment:  The authors should consider with regard to comment 3 to what extent these results are relevant to human embryonic development, since human embryos are not directly exposed as are chick embryos.

Our Response: The use of chicken embryo in the study is a very well-known model to study embryogenesis including embryotoxicity, which is why we opted to use it.

Reviewer comment: Since the majority of chick embryos did not survive 5 days of exposure, the authors should comment on the need for other experiments on embryonic responses at nonlethal doses.

Our Response: The sentence has been added to the conclusion as recommended (please refer to the Conclusion section) (page 8, lines 270-271).

Round 2

Reviewer 1 Report

The authors addressed all my comments.